# Shoot Organogenesis and Regeneration from Leaf Seedlings of *Diospyros oleifera* Cheng

**DOI:** 10.3390/plants12193507

**Published:** 2023-10-09

**Authors:** Yang Liu, Naifu Zhou, Chengrui Luo, Qi Zhang, Peng Sun, Jianmin Fu, Shuzhan Li, Ze Li

**Affiliations:** 1Key Laboratory of Cultivation and Protection for Non-Wood Forest Trees, Central South University of Forestry and Technology, Ministry of Education, Changsha 410004, China; 17797594548@163.com (Y.L.); luocr2l@163.com (C.L.); 2Hunan Provincial Key Laboratory of Forestry Biotechnology, College of Life Science and Technology, Changsha 410004, China; 3Research Institute of Non-Timber Forestry, Chinese Academy of Forestry, Zhengzhou 450003, China; zhangqi98572022@163.com (Q.Z.); sunpeng1017@163.com (P.S.); fjm371@163.com (J.F.); 4The Belt and Road International Union Research Center for Tropical Arid Non-Wood Forest in Hunan Province, Changsha 410004, China; 5The Key Laboratory of Non-Wood Forest Products of State Forestry Administration, Central South University of Forestry and Technology, Changsha 410004, China; 6Hunan Provincial Cooperative Center of Water Resources Research and Development, Changsha 410031, China

**Keywords:** adventitious shoots, rooting, *Callus induction*, disinfection, tissue culture

## Abstract

Persimmons (*Diospyros*) are economically important trees that are widely cultivated for wood production in China, and *Diospyros oleifera* Cheng is the main persimmon grafting stock. However, an efficient tissue culture system has not been perfected for *D*. *oleifera* due to the limits of proliferation and rooting cultures. Therefore, this study examined the effects of different plant growth regulators and concentrations on the primary culture of young embryos, induction of leaf callus, differentiation of adventitious shoots, and rooting culture of *D*. *oleifera*. The optimal formula for young embryo germination was 1/2 Murashige and Skoog (MS) medium containing 0.5 mg/L gibberellic acid (GA_3_); after 25 days, the sprouting rate of the young embryos was 67.3%. The best medium for leaf callus induction was 1/2MS medium containing 2.0 mg/L of 6-benzylaminopurine (6-BA) and 0.5 mg/L of naphthaleneacetic acid (NAA), and the callus induction rate was 88.9%. Then, the callus was transferred to 1/2MS medium containing 2.0 mg/L of zeatin (ZT), 0.5 mg/L of NAA, and 2.0 mg/L of thidiazuron (TDZ) to induce adventitious shoots; after 25 days, 5.4 buds were produced per explant, and the induction rate of the adventitious shoots was 88.3%. The adventitious shoots were transferred to 1/2MS medium containing 2.0 mg/L of ZT, 2.0 mg/L of 6-(γ,γ-dimethylallylamino)purine (2iP), and 0.1 mg/L of indole acetic acid (IAA) for the proliferation culture, for which the multiplication coefficient approached 7.5. After multiplication, the adventitious shoots were inoculated into 1/2MS medium containing 1.0 mg/L of indole butyric acid (IBA), 0.5 mg/L of NAA, and 1.0 mg/L of kinetin (KT); the rooting rate was 60.2%, and the average number of roots was 6.9.

## 1. Introduction

Persimmons (*Diospyros oleifera* Cheng) belong to the family Ebenaceae, which originated in East Asia. *Diospyros* is an important genus of traditional food tree species in China [1]. *D*. *oleifera* fruit can be eaten fresh or processed and is well loved. In addition, different parts of the persimmon with high medicinal value are commonly used in traditional Chinese medicine to reduce blood pressure, produce sobriety, cure throat pain, and so forth [2,3]. *D*. *oleifera* has good grafting affinity with sweet persimmon varieties such as ‘Taishuu’ and ‘Fuyu’, can cause tree dwarfing, and has early results, which makes it a good rootstock species [4].

*D. oleifera* is diploid (2n = 2x = 30) [5] and is the closest relative of hexaploid (2n = 6x = 90) persimmon [6]. *D*. *oleifera* is important for studying the origin of persimmon and is an important breeding parent. Compared to the diploid *D. lotus*, the sex type and fruit development characteristics of *D*. *oleifera* are more similar to those of hexaploid persimmon, so it has become the reference tree species for studying important persimmon traits, such as sex determination, fruit size, and fruit tannin synthesis. Most studies of *D*. *oleifera* focus on grafting affinity, plant regeneration technology, and morphological research, and a complete tissue culture system has not been reported [7]. Moreover, the marked browning that occurs during tissue culture has hindered research on the genetic transformation of *D*. *oleifera* [8].

*D. oleifera* usually takes the seedlings of seed germination as the stock, and takes the mother tree of more than 30 years as the scion. However, it is greatly affected by the season, so it cannot raise seedlings on a large scale. [9]. The establishment of a *D*. *oleifera* tissue culture and leaf regeneration system would not only accelerate seedling growth, but also enable molecular breeding work. Because the genes that control the flower sex in *D*. *oleifera* have been identified and have significant economic values, it is urgent to establish a genetic transformation system based on tissue culture [10].

During the process of persimmon tissue culture, graft browning, vitrification, and root difficulties often occur, particularly in *D. kaki* cv. Youho, *D*. *lotus*, and other persimmon varieties [11,12,13]. Browning in persimmon is mainly caused by phenolic substances [14]. In this study, we used young *D*. *oleifera* embryos as explants to establish a rapid *D*. *oleifera* tissue culture regeneration system through the sterile germination of young embryos, leaf regeneration, proliferation, and rooting culture. Our results lay the foundation for studies of genetic transformation and factory seedling cultivation.

## 2. Results

### 2.1. Explant Sterilization

The survival rates were significantly (*p* < 0.05) higher for the seeds that were disinfected with 75% ethanol for 2 min and 5% NaClO for 10 min than for the seeds that underwent other disinfectant treatments (Table 1). The germination rate of immature seeds in 75% ethanol for 2 min and 5% NaClO for 10 min was 46.0% higher than that for the seeds that underwent treatment with ethanol for 2 min and 0.1% HgCl_2_ for 6 min (*p* < 0.05). The seed contamination rate was lower after sterilization with 5% NaClO than with 0.1% HgCl_2_. The most suitable disinfection method for *D. oleifera* immature seeds was 75% ethanol for 2 min and then 5% NaClO for 10 min.

### 2.2. Seed Germination

As shown in Table 2, when the types and concentrations of plant growth regulators were fixed, the seed embryo germination rate was higher in the 1/2MS medium than in other media, particularly in 1/2MS + 0.5 mg/L GA_3_. The highest germination rate was 67.3%, which was 9.0% higher than in (1/2N) MS and 15.0% higher than in WPM (*p* < 0.05). The seed embryo germination rate decreased at lower and higher activated carbon concentrations. The germination rate with 1.0 mg/L of activated carbon added to 1/2MS + 0.5 mg/L GA_3_ medium was 17.1% higher than that without activated carbon and 12.7% higher than that with 2.0 mg/L activated carbon (*p* < 0.05). The medium that was suitable for germinating D. oleifera seed embryos was 1/2MS + 0.5 mg/L GA_3_ supplemented with 1.0 mg/L of activated carbon (Figure 1). After 30 days of secondary culture, the leaves were large and green, with excellent growth.

### 2.3. Leaf Disk Callus Induction

Leaf discs were inoculated into the culture media, as listed in Table 3. When the culture medium did not contain NAA, the callus formation rate was extremely low. When the NAA concentration was 0.5 mg/L, the callus formation rate was generally high. When the 6-BA concentration was 2.0 mg/L, the callus induction rate was 23.7% and 25.5% higher than with the 6-BA concentrations of 1.0 and 3.0 mg/L, respectively (*p* < 0.05). When the concentrations of the plant growth regulators were optimal, the leaf callus induction rate in the 1/2MS medium was 20.3% and 3.5% higher than that in the MS and (1/2N) MS media, respectively. As shown in Figure 2, the callus induction rate was the highest when 40 g/L of sucrose was added to 1/2MS + 2.0 mg/L 6-BA + 0.5 mg/L NAA medium. After 10 days of cultivation in the optimal medium formula, the calli were obtained from the leaf margin (Figure 3), and after 25 days of cultivation, a single leaf produced 1.5 × 1.5 × 1.5 cm^3^ of callus tissue. After culturing the callus tissue with this medium for 15 days, adventitious shoots appeared. After 25 days of adventitious shoot growth, the leaves were small and yellow.

### 2.4. Induction of Adventitious Shoots from Callus

As shown in Table 4 (Figure 4), adventitious shoots appeared in the transferred calli after 15 days of culture. The leaves gradually expanded by 25 days of culture. At 30 days, terminal buds began to elongate, and more leaves began to grow. Single buds were taller and larger at 40 days, with the optimal bud growth observed at 50 days. With 1.0 mg/L of TDZ and 0.5 mg/L of NAA, the adventitious shoot coefficient and adventitious shoot induction rate were low, and the bud growth was indefinite and weak. When the NAA concentration was kept constant, these values first increased and then decreased as the TDZ concentration increased. However, the ZT concentrations were higher than 2.0 mg/L in combination with TDZ, and ZT had a negative effect on the adventitious shoot formation. Callus inoculation into 1/2MS + 2.0 mg/L TDZ + 2.0 mg/L ZT + 0.5 mg/L NAA medium had the best adventitious shoot induction, with a coefficient of 5.4 and a bud induction rate of 83.3%.

### 2.5. Adventitious Shoot Proliferation in Culture

Individual adventitious shoots were cut from the callus and inoculated into the culture medium, as shown in Table 5 (Figure 5). In the hormone-free medium, the multiplication index remained at one. The medium supplemented only with 1.0 mg/L of ZT and 2iP produced multiple shoots, with a multiplication index of 1.9. When keeping the ZT concentration constant, the proliferation coefficient first increased and then decreased as the 2iP concentration increased. The greatest proliferation occurred in the medium containing 2.0 mg/L of ZT, 2.0 mg/L of 2iP, and 0.1 mg/L of IAA, and the proliferation coefficient was 7.5. In this medium, the shoots showed strong growth at 10 days. Interestingly, the multiplication index first increased and then decreased as the IAA concentration increased. The culture medium that was the most suitable for proliferation and culture was 1/2MS + 2.0 mg/L ZT + 2.0 mg/L 2iP + 0.1 mg/L IAA.

### 2.6. Rooting Culture

Individual adventitious shoots were cut from shoot clumps and inoculated into culture medium, as summarized in Table 6. The rooting rate and rooting number in the 1/2MS medium were significantly higher than in the other media. With the same basic medium, with an IBA concentration of 1.0 mg/L and an NAA concentration of 0.5 mg/L, the growth potential of the upper part of the root was very poor without adding KT but very strong after adding 1.0 mg/L of KT; when no NAA was added, the leaves turned yellow and curled up (Figure 6). As Figure 7 shows, when the sucrose concentration was 20 g/L, the rooting rate and average rooting number were significantly (*p* < 0.05) higher than with 10, 30, and 40 g/L; at 4.8%, 14.9%, and 48.7%; and 1.4, 3.2, and 5.0, respectively. The growth was the worst when the sucrose concentration was 40 g/L. The culture medium that was the most suitable for subsequent rooting was 1/2MS + 1.0 mg/L IBA + 1.0 mg/L KT + 0.5 mg/L NAA + 20 g/L sucrose.

## 3. Discussion

Although the MS medium is one of the most widely used basic media in woody plant tissue cultures, most studies of persimmon tissue cultures have shown that culture media with lower levels of trace elements (1/2MS and MS (1/2N)) are more suitable [15]. Consistent with previous experimental results, we found that the 1/2MS medium is more suitable for growing persimmons.

Cytokinin plays a key role in adventitious shoot regeneration, and ZT may best promote the growth and redifferentiation of persimmon explants [16,17]. Nakamura et al. compared the effects of 4-CPPU, ZT, and BA on the regeneration of the western Japanese persimmon stripe cotyls at different concentrations and found the highest rate of indefinite bud regeneration on modified MS medium supplemented with ZT 2.0 mg/L [18]. Sun et al. showed that ZT was effective for the regeneration of indefinite shoots, while BA was completely ineffective [19]. We compared the effects of ZT and BA on cotyl regeneration in persimmon and showed that at appropriate concentrations, both induced adventitious shoot regeneration, but ZT was more effective than BA. At 2.0 mg/L of ZT, the highest regeneration rate was 72.5%, and higher or lower ZT concentrations reduced regeneration. Different sugar concentrations have significant impacts on regeneration culture. In a study of persimmon-related tissue culture, it was found that when the sucrose concentration in the culture medium was high, the effect of inducing callus from leaves was better [20]. Therefore, in this study, the effect of different concentrations of sucrose on callus tissue was set. Finally, it was found that when the sucrose concentration was 40 g/L, the effect of inducing callus from leaves was the best, reaching 88.9%.

At present, there are no methods to proliferate and cultivate *D*. *oleifera*. The main methods for cultivating seedlings are the use of embryo germination and leaf regeneration cultures [4]. However, both methods have long cycles, it is difficult to control the growth of sterile seedlings, and severe browning occurs; this can greatly hinder the industrial cultivation of *D*. *oleifera* seedlings and the cultivation of stress-tolerant rootstocks. During proliferation culture, it is necessary to ensure the vigorous growth of sterile seedlings while maintaining a high proliferation coefficient and rate [21]. Therefore, we chose ZT and 2iP as the main cytokinins; both promote cell division and differentiation and the growth and development of active growth sites, and both stimulate plant cell division, thereby improving the germination rate of axillary buds of persimmon tissue culture seedlings. Due to the weak growth of proliferative clustered buds during proliferation culture, adding IAA can promote the growth rate of germinated axillary buds [22], thereby improving the overall proliferative effect. Finally, ZT, 2iP, and IAA were used as plant growth regulators in proliferation culture, resulting in a proliferation coefficient of 7.5.

The difficulty with plant rooting is related not only to the differences in varieties, but also to the type and concentration of hormones added to the medium and the rooting method selected. IBA and NAA are commonly used hormones in rooting culture [23,24]. IBA is mainly used to promote the growth of the taproot [25], while NAA promotes cell division and expansion and induces the formation of adventitious roots [26]. KT promotes cell division, induces bud differentiation and development, and delays leaf senescence [27]. In the initial research, we found that after rooting culture, the growth of the upper part of the root was weak, so a combination of IBA, KT, and NAA was chosen to induce adventitious roots. On the issue of auxin selection, Ai et al. [28] proved that IBA has a good effect on inducing the rooting of persimmon tissue culture seedlings, with a high survival rate of transplantation, due to the significant inhibitory effect of agonists on aging, particularly their ability to delay the destruction of the cellular structure [29]. As we previously found that the growth of tissue-cultured seedlings after rooting was extremely poor, with the main problem being the yellowing and curling of the leaves, we chose KT to improve growth after rooting. We found that adding 1.0 mg/L of IBA, 0.5 mg/L of NAA, and 1.0 mg/L of KT to the basic medium (1/2MS) had the best rooting effect, with an average of 6.9 roots; the growth was extremely vigorous. Sucrose is mainly used to provide a carbon source in the culture medium and is one of the indispensable components. We found that low concentrations of sucrose are more conducive to rooting culture during the rooting culture process, which is basically consistent with the research results of Li et al. [11]. Finally, the optimal sucrose concentration for rooting culture obtained in this study is 20 g/L.

There has been very little research on the tissue culture of *D*. *oleifera*, particularly proliferation culture. We developed a culture method with a proliferation coefficient of 7.5, providing a foundation for subsequent industrial seedling cultivation. However, much of the material produced weak and small adventitious shoots. We aim to improve this technology to use it for research on molecular breeding and to promote the development of stress-resistant rootstock culture.

## 4. Materials and Methods

### 4.1. Plant Materials

Wild *D*. *oleifera* (CNPC2009) fruits were picked in Xing’an County, Guangxi, China in the first 10 days of September 2020. The fruits were brought to the laboratory at room temperature, and the immature seeds were removed from the fruit, washed with water, shade-dried, and stored in a plastic bag at 4 °C in a refrigerator. *D*. *oleifera* as a rootstock has root-deep drought tolerance, strong growth potential, early results, and a dwarf tree shape [10].

### 4.2. Chemicals Used

The sources and catalogue numbers of the chemicals used in this study were as follows: 1/2MS (Solarbio, M8526); (1/2N)MS (Shanghai Earthquake Organism (verify this name), HZ1081–50 L); MS (Solarbio, M8520); WPM (Solarbio, LA6881); agar (Solarbio, A8190); sucrose (Hushi, 10021418); zeatin (ZT) (Yuanye, S18003); thidiazuron (TDZ) (Solarbio, T8050); 6-(γ,γ-dimethylallylamino)purine (2iP) (Solarbio, I8330); naphthaleneacetic acid (NAA) (Solarbio, N8010); indole acetic acid (IAA) (Solarbio, I8020); and kinetin (KT) (Solarbio, K8011).

### 4.3. Sterilization of Explants

The *D*. *oleifera* seeds stored in the refrigerator were soaked in water for at least 40 min, washed with detergent once or twice, and then washed with running water for at least 40 min. Then, they were disinfected with 75% alcohol for 1, 2, and 3 min on an ultra-clean worktable. The seeds were subjected to one of two treatments: they were soaked in 5% NaClO for 8, 10, and 12 min or in 0.1% HgCl_2_ for 5, 6, and 7 min. Finally, they were rinsed with sterile water three to five times. Then, the immature seed embryos were removed and inoculated into 1.0 mg/L GA_3_ medium. The contamination and germination rates were recorded after 25 days.

### 4.4. Seed Germination

Sterilized seed embryos were inoculated into 1/2 strength Murashige and Skoog (MS), (1/2N) MS, or woody plant medium (WPM). The basal media contained 0, 1.0, and 2.0 g/L activated carbon, respectively. These were treated with different plant growth regulators: 0.5 or 1.0 mg/L gibberellin (GA_3_) or 2.0 mg/L 6-benzylaminopurine (6-BA) + 0.5 mg/L naphthaleneacetic acid (NAA). The germination time of the seed embryos was determined, as was the germination rate after 20 days.

### 4.5. Leaf Disk Callus Induction

Leaves were removed from the germinated seed embryos, cut into 1 × 1 cm^2^ pieces, and inoculated into 1/2MS, (1/2N) MS, or MS medium for callus induction. The callus induction medium contained 6-BA (1.0, 2.0, or 3.0 mg/L) and NAA (0, 0.5, or 1.0 mg/L), and sucrose (35, 40, or 45 g/L) was added to the selected medium containing the optimal combination of plant growth regulators. The callus formation and browning rates were evaluated after 30 days.

### 4.6. Induction of Adventitious Shoots from Callus

For adventitious shoot induction, leaf-induced callus was cut into 1 × 1 × 1 cm^3^ pieces and inoculated into 1/2MS medium containing ZT (1.0, 2.0, or 3.0 mg/L), NAA (0.5, 1.0, 1.5, or 2.0 mg/L), and TDZ (1.0, 2.0, or 3.0 mg/L). The adventitious shoot induction rate and induction coefficient were evaluated after 25 days of culture.

### 4.7. Adventitious Shoot Proliferation in Culture

Adventitious shoots measuring about 2 cm were inoculated into medium containing zeatin (ZT) (1.0, 2.0, or 3.0 mg/L), 2iP (1.0, 2.0, or 3.0 mg/L), and indole-3-acetic acid (IAA) (0.1 mg/L). After 25 days of incubation, the growth and average plant height of the seedlings were assessed.

### 4.8. Rooting Cultivation

For rooting cultivation, the single adventitious shoots were transferred to 1/2MS, (1/2N) MS, MS, or WPM medium containing IBA (0.5, 1.0, or 1.5 mg/L), NAA (0.5 or 1.0 mg/L), and KT (1.0, 1.5, 2.0, or 2.5 mg/L); in addition, 10, 20, 30, and 40 g/L of sucrose was added to the culture medium. The roots and rooting rates were assessed after 30 days.

### 4.9. Culture Conditions

The culture medium without special instructions contained 30 g/L sucrose, 7 g/L agar, and a pH value of 5.5–5.8. The culture temperature was 28 ± 1 °C, the light intensity was 50–60 µmol/m^2^/s^1^, the light time was 14 h/day, and the medium was sterilized in a 121 °C autoclave cooker for 20 min.

### 4.10. Statistical Analysis

All experiments followed a completely randomized design with three replicates, and each treatment contained 30 explants. The data were analyzed using one-way analysis of variance, followed by Duncan’s multiple range test. The significance level was set at *p* < 0.05. Data are expressed as the mean ± standard error (SE). All statistical analyses were conducted using SPSS ver. 22.0 (IBM, Armonk, NY, USA).

The following were calculated: multiplication coefficient = increase in the index/the original number of individuals; induction coefficient = all callus-generated adventitious shoots/total callus number; inductivity = callus index/total number of callus-generated adventitious shoots; proliferation coefficient = multiplication number/original number of individuals; adventitious shoot coefficient = total number of callus-generated intimating buds/callus index; take root rate = number of adventitious shoots rooting/number of adventitious shoots inoculated into the culture medium.

## 5. Conclusions

The key to the in vitro propagation of *D. oleifera* is to determine the effects of different disinfection methods, basic culture media, plant growth regulators, and sucrose on the growth of tissue-cultured seedlings. According to the results of the current study, the optimal culture combination of the primary culture is 1/2MS + 0.5 mg/L GA_3_ + 30 g/L sucrose. The optimal medium for inducing callus from leaves is 1/2MS + 2.0 mg/L 6-BA + 0.5 mg/L NAA + 40 g/L sucrose. This medium formulation allows us to quickly obtain callus tissue. The optimal medium for inducing adventitious buds in callus tissue is 1/2 MS + 2.0 mg/L TDZ + 0.5 mg/L NAA + 2.0 mg/L ZT + 30 g/L sucrose. This medium formulation can obtain a large number of adventitious buds in a short period of time. The optimal medium formulation for proliferation culture is 1/2 MS + 2.0 mg/L ZT + 2.0 mg/L 2iP + 0.1 mg/L IAA + 30 g/L sucrose. This medium formulation not only proliferates more adventitious buds, but also exhibits excellent growth during the proliferation culture process. The rooting medium including 1/2 MS + 1.0 mg/L IBA + 1.0 mg/L KT + 0.5 mg/L NAA + 20 g/L sucrose produced the highest number of roots and the highest rooting rate. This tissue culture plan provides a complete and efficient *D. oleifera* tissue culture system, which can provide certain theoretical support for actual production. In addition, we also draw another conclusion that the establishment of a *D. oleifera* tissue culture system is not only related to plant growth regulators, but is also greatly influenced by the sucrose concentration and the type of basic culture medium.

## 6. Patents

The work reported here has a patent result that has been authorized. Ze Li, Yang Liu, Tao Zhang, Zi Yan Xu, Peng Sun, Jianmin Fu, Yini Mai, Shuzhan Li, Xiaohui Gao, a method for cultivating aseptic seedlings of *Diospyros oleifera* Cheng, 22 September 2022, China, CN202210762271.5.

## Figures and Tables

**Figure 1 plants-12-03507-f001:**
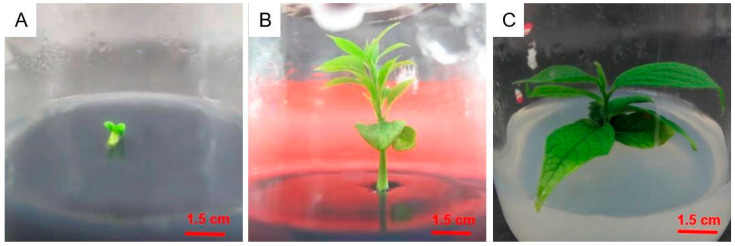
Germination of seed embryos of *Diospyros oleifera* Cheng. (**A**) Seed embryo germination after 10 days. (**B**) Seedling growth at 25 days. (**C**) Secondary culture at 30 days.

**Figure 2 plants-12-03507-f002:**
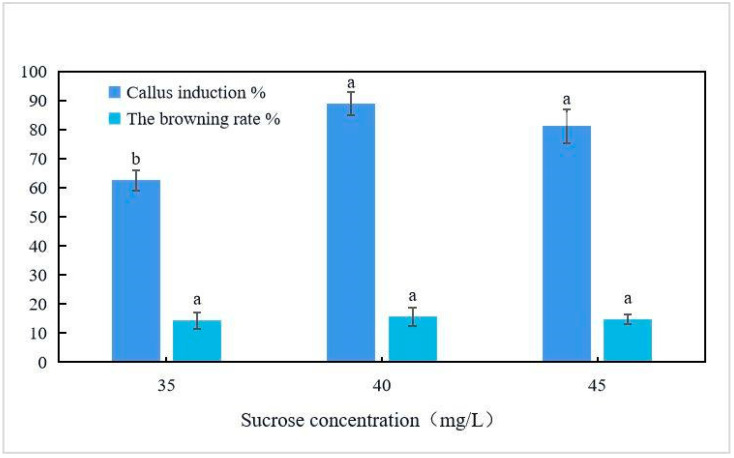
Effect of adding different concentrations of sucrose to 1/2MS + 2.0 mg/L 6-BA + 0.5 mg/L NAA medium on leaf callus induction. The same letters in rows are not significantly different at *p* ≤ 0.05.

**Figure 3 plants-12-03507-f003:**
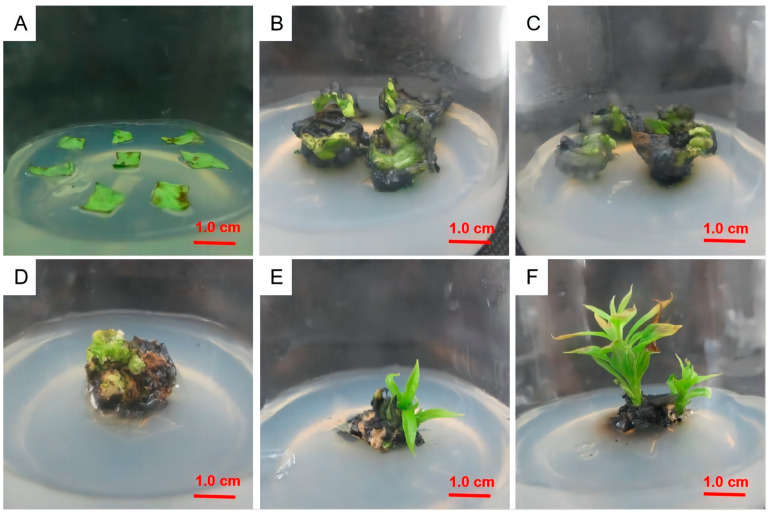
Leaf callus induction of *Diospyros oleifera* Cheng. (**A**) Leaf discs inoculated into culture medium. (**B**) Callus formed at the incision after 10 days of leaf culture. (**C**) Leaf discs were inoculated in culture medium for 20 days. (**D**) Complete callus tissue formed after 25 days. (**E**) Adventitious shoots after 15 days of callus culture. (**F**) Adventitious shoots after growth for 25 days.

**Figure 4 plants-12-03507-f004:**
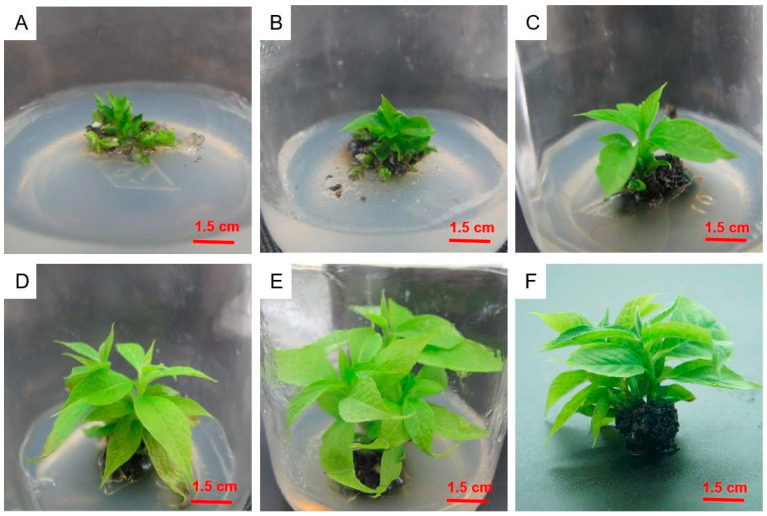
Adventitious shoots induced on the leaf callus. (**A**) Callus-generated adventitious shoots at 15 days. (**B**) After 25 days, the indefinite buds grew and the leaves unfolded; adventitious shoots grown for (**C**) 30, (**D**) 40, and (**E**,**F**) 50 days.

**Figure 5 plants-12-03507-f005:**
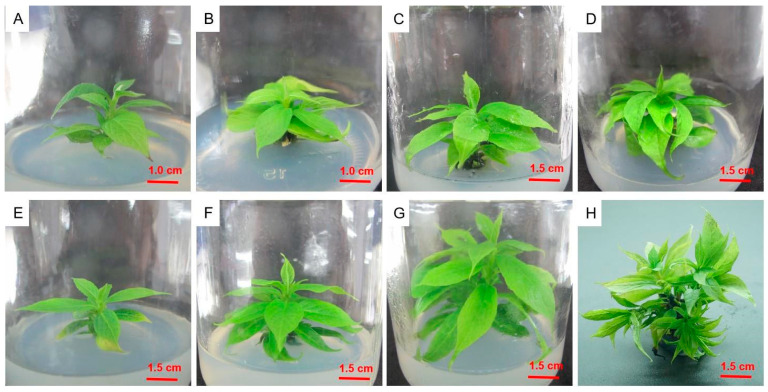
Adventitious shoot proliferation culture of *Diospyros oleifera* Cheng. (**A**–**D**) First-generation proliferation culture; (**E**,**F**) second-generation proliferation culture. (**A**,**E**) Single adventitious shoots were planted in the culture medium and cultured for 10 (**B**,**F**), 20 (**C**,**G**), and 30 (**D**,**H**) days.

**Figure 6 plants-12-03507-f006:**
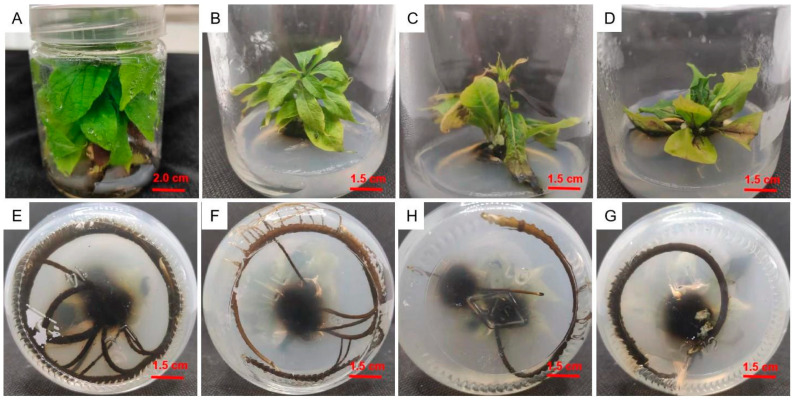
Growth of *D*. *oleifera* rooting culture under different plant growth regulators for 40 days in (**A**,**E**) 1/2MS + 1.0 mg/L IBA + 0.5 mg/L NAA + 1.0 mg/L KT; (**B**,**F**) 1/2MS + 1.0 mg/L IBA + 1.0 mg/L KT; (**C**,**H**) 1/2MS + 2.0 mg/L IBA; and (**D**,**G**) 1/2MS + 1.0 mg/L IBA + 0.5 mg/L NAA + 1.0 mg/L KT.

**Figure 7 plants-12-03507-f007:**
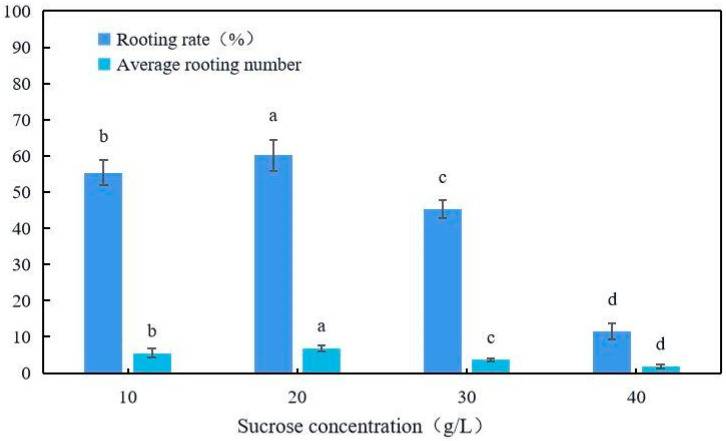
Rooting culture of *D*. *oleifera* in 1/2MS + 1.0 mg/L IBA + 0.5 mg/L NAA + 1.0 mg/L KT medium. The same letters in rows are not significantly different at *p* ≤ 0.05.

**Table 1 plants-12-03507-t001:** Effects of disinfectant types and times on disinfection of *D*. *oleifera* seeds.

75% Ethanol Min	5% NaClO Min	1% HgCl_2_ Min	Contamination Rate %	Germination Rate %
1	0	5	55.7 ± 1.2 a	27.6 ± 2.7 e
1	0	6	47.3 ± 5.3 b	28.5 ± 1.0 e
1	0	7	42.5 ± 8.3 b	30.1 ± 4.3 e
2	0	5	43.3 ± 4.0 b	38.6 ± 5.7 d
2	0	6	39.7 ± 6.4 bc	43.3 ± 6.3 c
2	0	7	35.6 ± 1.3 c	40.7 ± 7.7 cd
3	0	5	36.0 ± 0.7 c	42.3 ± 2.5 c
3	0	6	33.6 ± 1.3 c	37.3 ± 3.6 d
3	0	7	30.3 ± 2.6 cd	33.7 ± 4.6 d
1	8	0	43.3 ± 3.6 b	36.3 ± 5.7 d
1	10	0	40.7 ± 1.3 bc	38.3 ± 8.5 d
1	12	0	36.7 ± 8.6 c	40.7 ± 3.6 cd
2	8	0	28.3 ± 9.6 cd	53.7 ± 3.7 b
2	10	0	25.0 ± 1.0 d	63.5 ± 4.6 a
2	12	0	21.7 ± 4.6 d	54.3 ± 2.0 b
3	8	0	22.0 ± 1.3 d	39.3 ± 1.1 cd
3	10	0	18.9 ± 5.7 e	43.3 ± 1.2 c
3	12	0	13.3 ± 4.6 e	36.7 ± 3.7 d

Means followed by the same letters in rows are not significantly different at *p* ≤ 0.05.

**Table 2 plants-12-03507-t002:** Effects of different basal media and plant growth regulators on the germination of *Diospyros oleifera* Cheng seed embryos.

Medium	Additives	Activated Carbon g/L	Germination Rate %
1/2MS	2.0 6-BA + 0.5 NAA	1.0	51.2 ± 3.7 bc
1/2MS	0.5 GA_3_	1.0	67.3 ± 5.6 a
1/2MS	1.0 GA_3_	1.0	58.7 ± 8.0 b
(1/2N) MS	2.0 6-BA + 0.5 NAA	1.0	49.5 ± 6.6 c
(1/2N) MS	0.5 GA_3_	1.0	58.3 ± 6.3 b
(1/2N) MS	1.0 GA_3_	1.0	51.5 ± 7.5 bc
WPM	2.0 6-BA + 0.5 NAA	1.0	45.1 ± 8.7 c
WPM	0.5 GA_3_	1.0	52.3 ± 9.8 b
WPM	1.0 GA_3_	1.0	49.9 ± 6.4 bc
1/2MS	0.5 GA_3_	0	50.2 ± 3.3 bc
1/2MS	0.5 GA_3_	2.0	54.6 ± 4.3 b

Means followed by the same letters in rows are not significantly different at *p* ≤ 0.05.

**Table 3 plants-12-03507-t003:** Effects of different basal media and plant growth regulators on leaf callus formation of *Diospyros oleifera* Cheng.

Culture Medium	6-BA mg/L	NAA mg/L	Callus Induction Rate %	Browning Rate %
1/2MS	1.0	0	3.8 ± 0.6 g	8.0 ± 1.3 e
1/2MS	2.0	0	19.5 ± 2.7 f	10.3 ± 1.6 e
1/2MS	3.0	0	24.5 ± 2.6 e	21.3 ± 2.4 de
1/2MS	1.0	0.5	65.2 ± 3.7 b	16.2 ± 1.7 f
1/2MS	2.0	0.5	88.9 ± 4.6 a	15.7 ± 1.3 d
1/2MS	3.0	0.5	63.4 ± 2.6 b	11.5 ± 3.4 e
1/2MS	1.0	1	51.5 ± 3.6 d	25.1 ± 4.6 d
1/2MS	2.0	1	59.2 ± 2.7 c	28.4 ± 1.5 c
1/2MS	3.0	1	41.7 ± 2.7 d	35.3 ± 3.6 b
(1/2N) MS	2.0	0.5	85.4 ± 5.9 a	13.2 ± 2.1 d
MS	2.0	0.5	68.6 ± 4.7 b	9.9 ± 2.5 e

Means followed by the same letters in rows are not significantly different at *p* ≤ 0.05.

**Table 4 plants-12-03507-t004:** Effects of different plant growth regulator combinations on the induction of callus adventitious shoots of *D. oleifera* leaves.

TDZ mg/L	ZT mg/L	NAA mg/L	Adventitious Shoot Induction Rate %	Adventitious Shoot Coefficient	Growth Status of the Adventitious Shoots
1	1	0.5	20.4 ± 1.3 e	2.2 ± 1.3 c	Yellow and short leaves
1	2	0.5	32.4 ± 4.5 d	2.7 ± 1.5 b	Yellow leaves with withered tips
1	3	0.5	26.7 ± 3.3 e	1.4 ± 0.3 e	Leaf tips rarely withered and soft leaves
2	1	0.5	53.9 ± 1.5 b	3.0 ± 1.3 b	Growth more vigorous, but leaf base softer
2	2	0.5	83.3 ± 3.7 a	5.4 ± 2.1 a	Large and green leaves
2	3	0.5	47.2 ± 1.3 b	1.5 ± 0.2 de	Growth more vigorous, leaves yellowed
3	1	0.5	41.1 ± 2.3 bc	1.6 ± 0.4 d	Growth relatively vigorous, but leaf tip withered
3	2	0.5	34.6 ± 5.3 d	1.2 ± 1.1 e	Test tube seedlings short, and leaves withered
3	3	0.5	21.3 ± 3.2 e	1.1 ± 0.7 e	Seedlings short and leaves yellow
2	2	1	11.5 ± 2.2 f	1.7 ± 0.8 d	Weak growth with curled, yellowed leaves
2	2	1.5	10.4 ± 3.3 f	1.0 ± 0.5 e	Vigorous growth, but very low value-added coefficient
2	2	2	9.7 ± 2.3 f	1.0 ± 0.3 e	Seedlings short with withered leaf tips

Means followed by the same letters in rows are not significantly different at *p* ≤ 0.05.

**Table 5 plants-12-03507-t005:** Effects of different plant growth regulators on *D*. *oleifera* proliferation and culture.

ZT mg/L	2iP mg/L	IAA mg/L	Multiplication Coefficient	Material Growth Status
0	0	0	1.0	No proliferation
1	1	0	1.9 ± 0.33 e	Test tube seedlings short; leaves with black spots
1	2	0	2.4 ± 0.67 d	Seedlings taller, but leaves yellow and withered
1	3	0	1.7 ± 0.25 e	Short with thick, green, large leaves
2	1	0	3.1 ± 0.10 c	Tall but with withered leaf tips
2	2	0	5.5 ± 1.33 b	Taller with large, emerald green leaves
2	3	0	4.3 ± 0.33 c	Stem segments short; large yellow leaves
3	1	0	4.0 ± 0.21 cd	Stem sections short; large yellow withered leaves
3	2	0	4.2 ± 0.56 c	Seedlings taller with larger but yellow leaves
3	3	0	4.1 ± 0.43 cd	Seedlings taller with large, emerald green leaves
2	2	0.05	6.0 ± 0.52 b	Seedlings shorter with large, emerald green leaves
2	2	0.1	7.5 ± 0.33 a	Seedlings taller with large, emerald green leaves
2	3	0.15	2.5 ± 0.58 f	Seedlings shorter with large, emerald green leaves

Means followed by the same letters in rows are not significantly different at *p* ≤ 0.05.

**Table 6 plants-12-03507-t006:** Effects of different basal media and hormones on *D*. *oleifera* rooting culture.

Basal Culture Medium	IBA mg/L	NAA mg/L	KT mg/L	Rooting Rate %	Average Root Number
MS	0	0	0	0	0
1/2MS	0.5	0.5	1.0	33.4 ± 2.1 c	2.1 ± 0.2 d
(1/2N) MS	0.5	0.5	1.0	25.8 ± 3.9 d	1.4 ± 0.3 e
MS	0.5	0.5	1.0	17.5 ± 1.4 e	1.3 ± 0.5 e
WPM	0.5	0.5	1.0	16.2 ± 3.5 e	1.4 ± 0.6 e
1/2MS	1.0	0.5	1.0	60.2 ± 4.2 a	6.9 ± 2.8 a
(1/2N) MS	1.0	0.5	1.0	44.5 ± 2.3 b	4.4 ± 1.6 b
MS	1.0	0.5	1.0	38.1 ± 6.4 c	3.4 ± 1.4 c
WPM	1.0	0.5	1.0	28.4 ± 4.1 d	2.8 ± 1.5 d
1/2MS	1.5	0.5	1.0	44.6 ± 3.1 b	4.3 ± 0.5 b
(1/2N) MS	1.5	0.5	1.0	30.1 ± 2.6 c	3.1 ± 1.6 c
MS	1.5	0.5	1.0	23.2 ± 4.7 d	2.0 ± 0.6 d
WPM	1.5	0.5	1.0	22.1 ± 1.3 d	1.5 ± 0.5 e
1/2MS	1.0	1.0	1.5	10.4 ± 3.5 e	1.7 ± 1.1 e
1/2MS	1.0	0.5	2.0	21.0 ± 2.4 d	4.3 ± 2.4 b
1/2MS	1.0	0.5	2.5	18.5 ± 3.4 de	2.9 ± 1.2 d
1/2MS	1.0	0.5	0.5	41.4 ± 5.6 b	4.5 ± 0.9 b
1/2MS	1.0	0.5	0	33.4 ± 2.1 c	3.6 ± 1.2 c
1/2MS	1.0	0	1.0	22.5 ± 3.3 d	1.1 ± 0.2 e

Means followed by the same letters in rows are not significantly different at *p* ≤ 0.05.

## Data Availability

All data are available in the manuscript.

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
