# Peer review of "Shoot Organogenesis and Regeneration from Leaf Seedlings of Diospyros oleifera Cheng"

_plants, 2023, doi:10.3390/plants12193507_

Round 1
Reviewer 1 Report
Article: „Shoot Organogenesis and Regeneration from Sterile Seedling Leaves of Diospyros oleifera Cheng”
The authors presented adventitious shoot regeneration on leaf explants of wild Diospyros oleifera Cheng seedlings, among others, for mass production of rootstocks. However, they did not present the characteristics of the mother material, whether it had been previously tested and selected. It is know that seedling rootstocks of persimmons are highly heterozygous and exhibit considerable variation in vigour, which leads to some difficulties in orchard management. In addition, adventitious regeneration can also lead to variability. Please characterise the mother plants (CNPC2009), stability of seed progeny and explain why Authors used adventitious regeneration. As it stands, the introduction, purpose and conclusions are not consistent.
To achieve callus tissue-mediated regeneration, the authors used methods previously described for other Diospyros species. A novelty is the use of different sugar concentrations, but these experiments are sparsely described in M&M and Results, and omitted in the Discussion. Please clearly underline novelty and practical use of obtained results. Did Author obtained cyclic shoot formation?
Line 164 – instead “tufted buds” should be shoot clumps
Figure 6 – in the title please add the growth regulators used in the media containing different sucrose levels
Line 200-210, I suggest to delete this fragment. It is out of topic
Other comments are included in the text.

Author Response
Response to Reviewer 1 Comments
|
||
1. Summary |
|
|
Thank you for your letter and for the reviewers’ comments concerning our manuscript entitled,the topic by“Shoot Organogenesis and Regeneration from Sterile Seedling Leaves of Diospyros oleifera Cheng” be revised as “Shoot Organogenesis and Regeneration from Leaf Seedlings of Diospyros oleifera Cheng” (plants-2620836). Those comments are all valuable and very helpful for us to revise and improve our manuscript. We have studied those comments carefully and have made related modifications and corrections in our manuscript. We hope to meet with the approvals. All the modifications and corrections are displayed in the manuscript. The main corrections in the paper and the responds to the reviewer's comments are as flowing: |
||
2. Questions for General Evaluation |
Reviewer’s Evaluation |
Response and Revisions |
Does the introduction provide sufficient background and include all relevant references? |
Must be improved |
The introduction has been exhaustively modified |
Are all the cited references relevant to the research? |
Can be improved |
Missing references were added |
Is the research design appropriate? |
Can be improved |
Unjustified experimental design has been modified |
Are the methods adequately described? |
Must be improved |
Missing sucrose concentration and number of samples have been added |
Are the results clearly presented? |
Can be improved |
The conclusion section has been rewritten |
Are the conclusions supported by the results? |
Must be improved |
The conclusions have been modified |
3. Point-by-point response to Comments and Suggestions for Authors |
||
Comments 1: Please characterise the mother plants (CNPC2009), stability of seed progeny and explain why Authors used adventitious regeneration. |
||
Response 1: Thank you for pointing this out. The mother plants Diospyros oleifera Cheng (CNPC2009) Seed offspring are extremely stable. We used a total of 300 seeds during the study, and all seeds had the same traits as the mother strain after germination. In addition, in reference [10], the same conclusion can be drawn that stability of seed progeny is extremely stable. In the process of tissue culture, adventitious regeneration is a very critical part, and multiple indefinite buds developed from single cells can be obtained in the primary culture, which is completely consistent with the genotype of the mother and can retain the excellent traits of the mother strain to the greatest extent. |
||
Comments 2: As it stands, the introduction, purpose and conclusions are not consistent. |
||
Response 2: We have revised the paper extensively, especially in introduction and purpose and conclusions, to keep the three consistent. |
||
Comments 3:To achieve callus tissue-mediated regeneration, the authors used methods previously described for other Diospyros species. A novelty is the use of different sugar concentrations, but these experiments are sparsely described in M&M and Results, and omitted in the Discussion. Please clearly underline novelty and practical use of obtained results. Did Author obtained cyclic shoot formation? |
||
Response 3: We added a discussion on sucrose in the discussion section and compared it with previous studies. We performed obtained cyclic shoot formation during culture. |
||
Comments 4: Line 164 – instead “tufted buds” should be shoot clumps |
||
Response 4: Tufted buds be revised as shoot clumps. |
||
Comments 5: Figure 6 – in the title please add the growth regulators used in the media containing different sucrose levels |
||
Response 5: We added the used growth regulator to the sixth image caption. |
||
Comments 6: Line 200-210, I suggest to delete this fragment. It is out of topic |
||
Response 6: We have removed this part in the discussion. |
||
Comments 7: line-2 I suggest to change "Leaf Seedlings" |
||
Response 7: We have changed the title toShoot Organogenesis and Regeneration from Sterile Seedling Leaves of Diospyros oleifera Cheng |
||
Comments 8: line-19 ‘due to the limits of proliferation and rooting cultures’It is not clear |
||
Response 8: At the present stage, there are few studies on the tissue culture system of D. oleifera, and the growth plant is weak, so I wrote this sentence. |
||
Comments 9: line-34(The in vitro plant regeneration procedure described here should help improve D. oleifera Cheng through genetic engineering. )Out of topic. The Authors did't characterise the mother plant and what need modification. |
||
Response 9: We deleted the inappropriate sentence. |
||
Comments 10: line-41 What parts of the plants? |
||
Response 10: Specific graft material has been added. |
||
Comments 11: line-44 Please add the name of cultivars |
||
Response 11: The names of the specific varieties have been added. |
||
Comments 12: line-45 ‘which makes it a good rootstock species’ It is not clear |
||
Response12: This sentence refers to the grafting effect of D. oleifera and some plants is better, so it is a better rootstock. |
||
Comments 13: line-51 Please add References |
||
Response 13: ‘Most studies of D. oleifera focus on grafting affinity, plant regeneration technology, and morphological research.’ Specific references are added later to this sentence |
||
Comments 14: line-56 Please add what components are most commonly used, scions and rootstocks |
||
Response 14: Scions and rootstocks has been added to the text. |
||
Comments 15: line-57 ’so it is not realistic for industrial seed culture’ It is not clear |
||
Response 15: Due to the imperfect tissue culture system, seedlings can not be raised through the factory, so there are very few D. oleifera plants. In this study, a large number of tissue culture seedlings can be obtained through proliferation, so as to promote a large number of D. oleifera seedlings with excellent traits. |
||
Comments 16: line-59 Please explain the relationship more clearly(Because the genes that control specific economically valuable traits in D. oleifera have been identified, the establishment of an efficient tissue culture system is imminent.) |
||
Response 16: We have modified this section to read ‘Because the genes that control flower sex in D. oleifera have been identified and have significant economic value, it is urgent to establish a genetic transformation system based on tissue culture.’ |
||
Comments 17: line-106 (As shown in Figure 2, leaf induction from callus was best when the sucrose concentrati on was 40 g/L.) It is not clear, what growth regulators were used together with 40 g/L sucrose. |
||
Response 17: We have modified this section to read ‘As shown in Figure 2, the callus induction rate was highest when 40g/L sucrose was added to 1/2MS+2.0mg/L 6-BA+0.5mg/L NAA medium.’ |
||
Comments 18: line-116 on medium containing ....please add growth regulators used |
||
Response 18: Relevant growth regulators have been added. |
||
Comments 19: line-125 Bud spots is not precise statement |
||
Response 19: We modified the inaccurate representation and modified bud spots to adventitious shoots. |
||
Comments 20: line-140 I suggest: Adventitious shoots induced on the leaf callus |
||
Response 20: Callus-generated adventitious shoots of Diospyros oleifera Cheng. be revised as Adventitious shoots induced on the leaf callus |
||
Comments 21: line-252 Please provide characteristics as a rootstock. |
||
Response 21: Features as a rootstock were added in the material section.D. oleifera as a rootstock root deep drought tolerance, strong growth potential, early results, tree shape dwarf. |
||
Comments 22: line-273/291/296 Please include what concentration of the sucrose was used. |
||
Response 22: Parts without annotated sucrose concentration, all used 30g / L sucrose, and a description was added in section 4.9. |
||
Comments 23: line-284 In every treatment with growth regulators? |
||
Response 23: Different concentrations of sucrose were used in the optimal medium. |
||
Comments 24: line-318 the adventitious shoots that grew from callus were cut into single buds, and added‘ is recommended to modify it to ’the single adventitious shoots were transferred’. |
||
Response 24: the adventitious shoots that grew from callus were cut into single buds, and added‘ be revised as ’the single adventitious shoots were transferred’. |
||
4. Response to Comments on the Quality of English Language |
||
The English in this document has been checked by at least two professional editors, both native speakers of English. For a certificate, please see: http://www.textcheck.com/certificate/B6GfIn The above statement is here to inform reviewers—who may not be native speakers of English—that the English in this document has been professionally checked. If the link to the certificate above is deleted and copied into a letter then the reviewers will not see it. |

Reviewer 2 Report
The manuscript “Shoot organogenesis and regeneration from sterile seedling leaves of Diospyros oleifera Cheng” discusses a detailed protocol for the regeneration of persimmon plants from leaf explants.
The authors suggest that these experiments may be useful in the future for producing transgenic persimmon plants. The manuscript is compiled according to the rules of the journal and contains the necessary sections.
However, revision of the manuscript is required. The work does not provide methodological approaches that consider the consequences and selection when obtaining transgenic plants, if the authors state that this protocol could be useful in genetic transformation. For example, the reaction of the formation of regenerants and plant growth on the standard antibiotic kanamycin and/or other methods was not assessed. Also, the work did not analyze different types of explants that differ in age or origin. For example, very small leaves, unfolded leaves, petioles, stem explants. This comparative approach is standard for this type of research.This is not covered in the Introduction. For the given histograms and tables, the explanation does not provide statistical analysis. It is unknown how many explants the authors used when conducting the experiments in subsections 2.1; 2.2; 2.3; 2.4; 2.5. There are percentages, but no number of explants. Therefore, it is difficult to assess the reliability of the experiments performed.
In experiments with browning of explants, the authors did not study the possibility of using compounds that relieve this effect, for example, some antioxidants.This also needed to be discussed in the Introduction or Discussion. Section 4.3 does not clearly state the age of the seeds used. Why do the authors write that young embryos? Were the seeds unmature?
Minor notes - sources must be enclosed in square brackets.
In general, the manuscript requires a major revision.
Author Response
Response to Reviewer 2 Comments
|
||
1. Summary |
|
|
Thank you for your letter and for the reviewers’ comments concerning our manuscript entitled,the topic by“Shoot Organogenesis and Regeneration from Sterile Seedling Leaves of Diospyros oleifera Cheng” be revised as “Shoot Organogenesis and Regeneration from Leaf Seedlings of Diospyros oleifera Cheng” (plants-2620836). Those comments are all valuable and very helpful for us to revise and improve our manuscript. We have studied those comments carefully and have made related modifications and corrections in our manuscript. We hope to meet with the approvals. All the modifications and corrections are displayed in the manuscript. The main corrections in the paper and the responds to the reviewer's comments are as flowing: |
||
2. Questions for General Evaluation |
Reviewer’s Evaluation |
Response and Revisions |
Does the introduction provide sufficient background and include all relevant references? |
Must be improved |
The introduction has been exhaustively modified |
Are all the cited references relevant to the research? |
Can be improved |
Missing references were added |
Is the research design appropriate? |
Yes |
Unjustified experimental design has been modified |
Are the methods adequately described? |
Must be improved |
Missing sucrose concentration and number of samples have been added |
Are the results clearly presented? |
Can be improved |
The conclusion section has been rewritten |
Are the conclusions supported by the results? |
Can be improved |
The conclusions have been modified |
3. Point-by-point response to Comments and Suggestions for Authors |
||
Comments 1: The work does not provide methodological approaches that consider the consequences and selection when obtaining transgenic plants, if the authors state that this protocol could be useful in genetic transformation. lease characterise the mother plants (CNPC2009), stability of seed progeny and explain why Authors used adventitious regeneration.For example, the reaction of the formation of regenerants and plant growth on the standard antibiotic kanamycin and/or other methods was not assessed. |
||
Response 1: Thank you for pointing this out. We did not study the effect of antibiotics in this study, and the main reason is that the system is already used in genetic transformation,(Mai, Y.N.; Liu, Y.; Yuan, J.Y.; et al. "Establishment of an efficient genetic transformation system: A case study of RNAi-induced silencing of the transcription factor MeGI in Diospyros oleifera Cheng seedlings." [J]. Scientia Horticulturae 308. (2023). DOI: 10.1016/J.SCIENT A.2022.111560)this purpose is to show the complete tissue culture system of the persimmon to see if it can be used in other genes. |
||
Comments 2: Also, the work did not analyze different types of explants that differ in age or origin. For example, very small leaves, unfolded leaves, petioles, stem explants. This comparative approach is standard for this type of research.This is not covered in the Introduction. |
||
Response 2: Immature seeds were chosen because there is a thick seed coat in the outer epidermis of mature seeds, which leads to extremely low germination rates. Because there was a dense layer of villi in the surface layer of the persimmon, which caused great difficulty in disinfection and low survival rate, the immature seeds were finally used as explants. |
||
Comments 3:For the given histograms and tables, the explanation does not provide statistical analysis. It is unknown how many explants the authors used when conducting the experiments in subsections 2.1; 2.2; 2.3; 2.4; 2.5. There are percentages, but no number of explants. Therefore, it is difficult to assess the reliability of the experiments performed. |
||
Response 3: We added specific statistics in part 4, and in part 4.10, we explained the methods of data analysis in detail and the number of statistics. |
||
Comments 4: In experiments with browning of explants, the authors did not study the possibility of using compounds that relieve this effect, for example, some antioxidants.This also needed to be discussed in the Introduction or Discussion. Section 4.3 does not clearly state the age of the seeds used. Why do the authors write that young embryos? Were the seeds unmature? |
||
Response 4: In experiments with browning of explants,since the previous study found that the effect of leaf induction callus was not significantly different after the addition of antifusonizer, it was not filled in.The explants we used were immature seeds, so described as young embryos at 4.3. |
||
Comments 5: Minor notes - sources must be enclosed in square brackets. |
||
Response 5: We have enclosed the references in the text in square brackets. |
||
Response 6: We have removed this part in the discussion. |
||
Comments 7: line-2 I suggest to change "Leaf Seedlings" |
||
Response 7: Thank you very much for your rigorous academic attitude and suggestion.We have changed the title toShoot Organogenesis and Regeneration from Sterile Seedling Leaves of Diospyros oleifera Cheng |
||
Comments 8: line-19 ‘due to the limits of proliferation and rooting cultures’It is not clear |
||
Response 8: At the present stage, there are few studies on the tissue culture system of D. oleifera, and the growth plant is weak, so I wrote this sentence. |
||
Comments 9: line-34(The in vitro plant regeneration procedure described here should help improve D. oleifera Cheng through genetic engineering. )Out of topic. The Authors did't characterise the mother plant and what need modification. |
||
Response 9: We deleted the inappropriate sentence. |
||
Comments 10: line-41 What parts of the plants? |
||
Response 10: Specific graft material has been added. |
||
Comments 11: line-44 Please add the name of cultivars |
||
Response 11: The names of the specific varieties have been added. |
||
Comments 12: line-45 ‘which makes it a good rootstock species’ It is not clear |
||
Response12: This sentence refers to the grafting effect of D. oleifera and some plants is better, so it is a better rootstock. |
||
Comments 13: line-51 Please add References |
||
Response 13: ‘Most studies of D. oleifera focus on grafting affinity, plant regeneration technology, and morphological research.’ Specific references are added later to this sentence |
||
Comments 14: line-56 Please add what components are most commonly used, scions and rootstocks |
||
Response 14: Scions and rootstocks has been added to the text. |
||
Comments 15: line-57 ’so it is not realistic for industrial seed culture’ It is not clear |
||
Response 15: Due to the imperfect tissue culture system, seedlings can not be raised through the factory, so there are very few D. oleifera plants. In this study, a large number of tissue culture seedlings can be obtained through proliferation, so as to promote a large number of D. oleifera seedlings with excellent traits. |
||
Comments 16: line-59 Please explain the relationship more clearly(Because the genes that control specific economically valuable traits in D. oleifera have been identified, the establishment of an efficient tissue culture system is imminent.) |
||
Response 16: We have modified this section to read ‘Because the genes that control flower sex in D. oleifera have been identified and have significant economic value, it is urgent to establish a genetic transformation system based on tissue culture.’ |
||
Comments 17: line-106 (As shown in Figure 2, leaf induction from callus was best when the sucrose concentrati on was 40 g/L.) It is not clear, what growth regulators were used together with 40 g/L sucrose. |
||
Response 17: We have modified this section to read ‘As shown in Figure 2, the callus induction rate was highest when 40g/L sucrose was added to 1/2MS+2.0mg/L 6-BA+0.5mg/L NAA medium.’ |
||
Comments 18: line-116 on medium containing ....please add growth regulators used |
||
Response 18: Relevant growth regulators have been added. |
||
Comments 19: line-125 Bud spots is not precise statement |
||
Response 19: We modified the inaccurate representation and modified bud spots to adventitious shoots. |
||
Comments 20: line-140 I suggest: Adventitious shoots induced on the leaf callus |
||
Response 20: Callus-generated adventitious shoots of Diospyros oleifera Cheng. be revised as Adventitious shoots induced on the leaf callus |
||
Comments 21: line-252 Please provide characteristics as a rootstock. |
||
Response 21: Features as a rootstock were added in the material section.D. oleifera as a rootstock root deep drought tolerance, strong growth potential, early results, tree shape dwarf. |
||
Comments 22: line-273/291/296 Please include what concentration of the sucrose was used. |
||
Response 22: Parts without annotated sucrose concentration, all used 30g / L sucrose, and a description was added in section 4.9. |
||
Comments 23: line-284 In every treatment with growth regulators? |
||
Response 23: Different concentrations of sucrose were used in the optimal medium. |
||
Comments 24: line-318 the adventitious shoots that grew from callus were cut into single buds, and added‘ is recommended to modify it to ’the single adventitious shoots were transferred’. |
||
Response 24: the adventitious shoots that grew from callus were cut into single buds, and added‘ be revised as ’the single adventitious shoots were transferred’. |
||
4. Response to Comments on the Quality of English Language |
||
The English in this document has been checked by at least two professional editors, both native speakers of English. For a certificate, please see: http://www.textcheck.com/certificate/B6GfIn The above statement is here to inform reviewers—who may not be native speakers of English—that the English in this document has been professionally checked. If the link to the certificate above is deleted and copied into a letter then the reviewers will not see it. |

Round 2
Reviewer 1 Report
The Authors have made a lot of valuable changes. However, I have some comments, which I have included in the attached file.

Author Response
Response to Reviewer 1 Comments
|
||
1. Summary |
|
|
Thank you for your letter and for the reviewers’ comments concerning our manuscript entitled,the topic by “Shoot Organogenesis and Regeneration from Leaf Seedlings of Diospyros oleifera Cheng” (plants-2620836). Those comments are all valuable and very helpful for us to revise and improve our manuscript. We have studied those comments carefully and have made related modifications and corrections in our manuscript. We hope to meet with the approvals. All the modifications and corrections are displayed in the manuscript. The main corrections in the paper and the responds to the reviewer's comments are as flowing: |
||
2. Questions for General Evaluation |
Reviewer’s Evaluation |
Response and Revisions |
Does the introduction provide sufficient background and include all relevant references? |
Yes |
|
Are all the cited references relevant to the research? |
Can be improved |
Missing references were added |
Is the research design appropriate? |
Yes |
|
Are the methods adequately described? |
Can be improved |
Missing sucrose concentration and number of samples have been added |
Are the results clearly presented? |
Can be improved |
The conclusion section has been rewritten |
Are the conclusions supported by the results? |
Must be improved |
The conclusions have been modified |
3. Point-by-point response to Comments and Suggestions for Authors |
||
Comments 1: Line 56-58, To long sentence ‘The main method for reproducing D. oleifera is seeding grafting, commonly used rootstocks are seedlings with germinating seeds, and scions are mainly mother trees over 30 years old, but it is greatly affected by the season and the breeding cycle is long, so it is not realistic for industrial seed culture.’ |
||
Response 1: Thank you for pointing this out. We have shortened this sentence. In order to ensure the accuracy of its expression, we have retained part of the content. I hope you can agree to our revision. The modified sentence is as follows: D. oleifera usually takes the seedlings of seed germination as the stock stock, and the mother tree of more than 30 years as the scion. However, it is greatly affected by the season, so it cannot raise seedlings on a large scale. |
||
Comments 2: Line 140, Please change the font ‘Table 4. Effects of different plant growth regulator combinations on the induction of callus adventitious shoots of D. oleifera leaves.’ |
||
Response 2: Thank you very much for your rigorous academic attitude and suggestion. We have changed the format marked on the Plants template with font Palatino Linotype and size 5. |
||
Comments 3:Line 238-245, It is not clear. If the Authors suggest the link between inhibited growth of seedling and sucrose concentration, please add References and explanation. |
||
Response 3: Your question is very valuable, this conclusion should be concluded after the paragraph 'Sucrose is mainly used to provide a carbon source in the culture medium and is one of the indispensable components. We found that low concentrations of sucrose are more conducive to rooting culture during the rooting culture process, which is basically consistent with the research results of Li et al. [11].', thank you again for your reminder. |
||
Comments 4: Line 164, It is not clear. ‘D. oleifera as a rootstock root deep drought tolerance, strong growth potential, early results, tree shape dwarf.’ |
||
Response 4: First of all, thank you for your comments, we have added references after this sentence, so that we can clearly know the meaning and source of this sentence,thank you again for your reminder. |
||
Comments 5: Line 301-304, Please add information on sucrose concentration.‘482-484’ For rooting cultivation, the single adventitious shoots were transferred to 1/2MS, (1/2N) MS, MS, or WPM medium containing IBA (0.5, 1.0, or 1.5 mg/L), NAA (0.5 or 1.0 mg/L), and KT (1.0, 1.5, 2.0, or 2.5 mg/L). The roots and rooting rates were assessed after 30 days. |
||
Response 5: First of all, thank you for your comments, We added sucrose information to the new article, as follows. For rooting cultivation, the single adventitious shoots were transferred to 1/2MS, (1/2N) MS, MS, or WPM medium containing IBA (0.5, 1.0, or 1.5 mg/L), NAA (0.5 or 1.0 mg/L), and KT (1.0, 1.5, 2.0, or 2.5 mg/L), in addition, add 10, 20, 30, and 40g/L of sucrose to the culture medium.The roots and rooting rates were assessed after 30 days. |
||
Comments 6: Line 323, These are not conclusions, but an summary which is a repetition of the abstract. Hence, it requires correction. |
||
Response 6: First of all, thank you for your evaluation. We have revised the conclusion section to mainly refer to a few articles, which are all from Plants journals. [1] Abdelghaffar, A.M.; Soliman, S.S.; Ismail, T.A.; Alzohairy, A.M.; Latef, A.A.H.A.; Alharbi, K.; Al-Khayri, J.M.; Aljuwayzi, N.I.M.; El-Moneim, D.A.; Hassanin, A.A. In Vitro Propagation of Three Date Palm (Phoenix dactylifera L.) Varieties Using Immature Female Inflorescences. Plants 2023, 12, 644. https://doi.org/10.3390/plants12030644 [2] Yaroshko, O.; Pasternak, T.; Larriba, E.; Pérez-Pérez, J.M. Optimization of Callus Induction and Shoot Regeneration from Tomato Cotyledon Explants. Plants 2023, 12, 2942. https://doi.org/10.3390/plants12162942 [3] Fu, Y.; Shu, L.; Li, H.; Zhang, X.; Liu, X.; Ou, Z.; Liang, X.; Qi, X.; Yang, L. Establishment of Highly Efficient Plant Regeneration, Callus Transformation and Analysis of Botrytis cinerea-Responsive PR Promoters in Lilium brownii var. viridulum. Plants 2023, 12, 1992. https://doi.org/10.3390/plants12101992 |
Reviewer 2 Report
The manuscript Shoot Organogenesis and Regeneration from Sterile Seedling Leaves of Diospyros oleifera Cheng by Yang Liu, Naifu Zhou, Cheng rui Luo, Qi Zhang, Peng Sun, Jian min Fu, Shu zhan Li, Ze Li has been significantly revised.
Comments and corrections have been made.
The manuscript may be accepted for publication.
Author Response
Thank you for your letter and for the reviewers’ comments concerning our manuscript entitled,the topic by “Shoot Organogenesis and Regeneration from Leaf Seedlings of Diospyros oleifera Cheng” (plants-2620836). I sincerely thank you for your affirmation of our research. In addition, I will also send you the comments of another reviewer and the relevant modifications we have made. I hope to receive your affirmation.
3. Point-by-point response to Comments and Suggestions for Authors |
Comments 1: Line 56-58, To long sentence ‘The main method for reproducing D. oleifera is seeding grafting, commonly used rootstocks are seedlings with germinating seeds, and scions are mainly mother trees over 30 years old, but it is greatly affected by the season and the breeding cycle is long, so it is not realistic for industrial seed culture.’ |
Response 1: Thank you for pointing this out. We have shortened this sentence. In order to ensure the accuracy of its expression, we have retained part of the content. I hope you can agree to our revision. The modified sentence is as follows: D. oleifera usually takes the seedlings of seed germination as the stock stock, and the mother tree of more than 30 years as the scion. However, it is greatly affected by the season, so it cannot raise seedlings on a large scale. |
Comments 2: Line 140, Please change the font ‘Table 4. Effects of different plant growth regulator combinations on the induction of callus adventitious shoots of D. oleifera leaves.’ |
Response 2: Thank you very much for your rigorous academic attitude and suggestion. We have changed the format marked on the Plants template with font Palatino Linotype and size 5. |
Comments 3:Line 238-245, It is not clear. If the Authors suggest the link between inhibited growth of seedling and sucrose concentration, please add References and explanation. |
Response 3: Your question is very valuable, this conclusion should be concluded after the paragraph 'Sucrose is mainly used to provide a carbon source in the culture medium and is one of the indispensable components. We found that low concentrations of sucrose are more conducive to rooting culture during the rooting culture process, which is basically consistent with the research results of Li et al. [11].', thank you again for your reminder. |
Comments 4: Line 164, It is not clear. ‘D. oleifera as a rootstock root deep drought tolerance, strong growth potential, early results, tree shape dwarf.’ |
Response 4: First of all, thank you for your comments, we have added references after this sentence, so that we can clearly know the meaning and source of this sentence,thank you again for your reminder. |
Comments 5: Line 301-304, Please add information on sucrose concentration.‘482-484’ For rooting cultivation, the single adventitious shoots were transferred to 1/2MS, (1/2N) MS, MS, or WPM medium containing IBA (0.5, 1.0, or 1.5 mg/L), NAA (0.5 or 1.0 mg/L), and KT (1.0, 1.5, 2.0, or 2.5 mg/L). The roots and rooting rates were assessed after 30 days. |
Response 5: First of all, thank you for your comments, We added sucrose information to the new article, as follows. For rooting cultivation, the single adventitious shoots were transferred to 1/2MS, (1/2N) MS, MS, or WPM medium containing IBA (0.5, 1.0, or 1.5 mg/L), NAA (0.5 or 1.0 mg/L), and KT (1.0, 1.5, 2.0, or 2.5 mg/L), in addition, add 10, 20, 30, and 40g/L of sucrose to the culture medium.The roots and rooting rates were assessed after 30 days. |
Comments 6: Line 323, These are not conclusions, but an summary which is a repetition of the abstract. Hence, it requires correction. |
Response 6: First of all, thank you for your evaluation. We have revised the conclusion section to mainly refer to a few articles, which are all from Plants journals. [1] Abdelghaffar, A.M.; Soliman, S.S.; Ismail, T.A.; Alzohairy, A.M.; Latef, A.A.H.A.; Alharbi, K.; Al-Khayri, J.M.; Aljuwayzi, N.I.M.; El-Moneim, D.A.; Hassanin, A.A. In Vitro Propagation of Three Date Palm (Phoenix dactylifera L.) Varieties Using Immature Female Inflorescences. Plants 2023, 12, 644. https://doi.org/10.3390/plants12030644 [2] Yaroshko, O.; Pasternak, T.; Larriba, E.; Pérez-Pérez, J.M. Optimization of Callus Induction and Shoot Regeneration from Tomato Cotyledon Explants. Plants 2023, 12, 2942. https://doi.org/10.3390/plants12162942 [3] Fu, Y.; Shu, L.; Li, H.; Zhang, X.; Liu, X.; Ou, Z.; Liang, X.; Qi, X.; Yang, L. Establishment of Highly Efficient Plant Regeneration, Callus Transformation and Analysis of Botrytis cinerea-Responsive PR Promoters in Lilium brownii var. viridulum. Plants 2023, 12, 1992. https://doi.org/10.3390/plants12101992 |